# ENABLING REASONING LANGUAGE MODELS TO REVEAL THEIR TRUE THOUGHTS VIA CoT INVERSION

## ABSTRACT

Chain-of-Thought (CoT) enables large language models (LLMs) to tackle complex reasoning tasks by generating intermediate steps. Although CoT provides opportunities for improved interpretability and facilitates the monitoring of AI safety, the consistency between generated CoT and the model's actual reasoning process is not guaranteed. Models can output seemingly reasonable CoT that fails to reflect the true computational trajectory leading to the final answer. IIn this work, we introduce a novel approach called CoT Inversion to evaluate CoT faithfulness. We cast the problem in a probabilistic framework, viewing the genuine reasoning chain as a latent variable that mediates between the input and the answer. Leveraging variational inference with a scoring function, we infer this hidden CoT by effectively reversing the model's answer generation process; under our chosen variational family, the optimization reduces to an instance of the Expectation Maximization (EM) algorithm. Furthermore, we propose an explicit alignment objective that promotes similarity between the inferred latent CoT and the model's directly generated CoT, considering the explicit CoT as an informative and possibly unfaithful signal. Our approach enables the quantitative assessment of agreement between articulated and inferred reasoning processes, offering a practical metric of CoT faithfulness and strengthening our ability to interpret and trust the reasoning of language models.

## 1 INTRODUCTION

Reasoning Large Language Models (RLLMs) have demonstrated extraordinary abilities in a variety of reasoning-intensive tasks Guo et al. (2025); OpenAI (2024). Their performance is especially impressive when reasoning is used, which provides explicit, stepwise explanations for intermediate reasoning processes. The significance of CoT reasoning lies not only in its ability to boost task accuracy but also in its potential to illuminate the model's internal logic. By generating these intermediate steps, RLLMs invite external scrutiny, facilitating deeper understanding, systematic auditing, and ideally, it offering pathways for interpretability and safety improvements. CoT reasoning thus emerges as a powerful instrument for transparency, fostering user trust, and supporting the scientific evaluation of model decisions in a manner inaccessible to models that do not expose their intermediate steps.

However, the increasing reliance on CoT reasoning for safety and interpretability introduces vulnerabilities of its own. First, proactive monitoring of generated reasoning traces is widely proposed as a solution for identifying unfaithful or problematic model behaviors. Previous research, as illustrated in panel A of Figure 1, typically leverages monitoring mechanisms over the externally presented CoT to flag unfaithful explanations. Despite the practicality of this approach, recent studies reveal a fundamental shortcoming: the externally generated CoT does not always faithfully correspond to the model's real underlying inferential process. In other words, RLLMs are capable of constructing stepwise explanations that, while plausible and well-formed, do not reflect the causal or mechanistic pathway actually traversed to produce the answer Chua & Evans (2025); Baker et al. (2025); Chen et al. (2025b). Such post-hoc rationalizations may mislead researchers or practitioners into certifying model behaviors as "faithful" when, in fact, the linkage between the explanation and actual computation is spurious or absent Zhang et al. (2025); Turpin et al. (2023); Arcuschin et al. (2025). This discrepancy calls into question the dependability of current CoT monitoring methods and presents a serious obstacle for robust AI safety assessment.

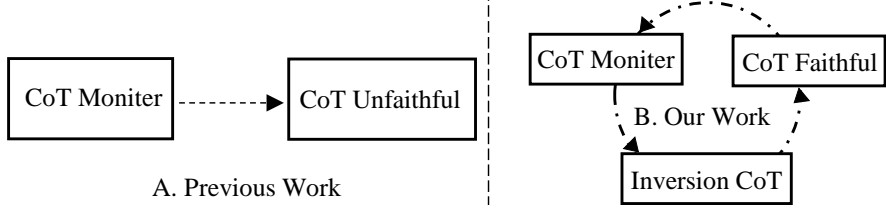

Figure 1: Workflow Comparison

The pressing need to identify faithful reasoning highlights the necessity of recovering the model's actual computational trajectory. As panel B of Figure 1 illustrates, our approach goes beyond accepting reported CoTs, emphasizing the inversion of internal reasoning to accurately evaluate CoT faithfulness. This distinction enables more reliable interpretability and safety assessment for RLLMs. A key research question emerges: How can we reconstruct the true internal reasoning path behind an RLLM's answer?

To tackle this problem, we propose a probabilistic modeling framework in which the true reasoning sequence is treated as a latent variable linking the input to the final answer. Specifically, we conceptualize the causal mechanism as flowing from input through an intermediate reasoning sequence to the answer. Given both the model's input and its generated answer, our objective is to invert this process, inferring the most plausible latent reasoning path responsible for the observed output. We operationalize this by modeling the posterior distribution over possible reasoning paths conditioned on input and answer. Due to the complexity and high dimensionality of textual reasoning traces, computing this posterior exactly is generally intractable. To render inference tractable, we apply variational inference, introducing a dedicated inference model to efficiently approximate the posterior over latent reasoning paths. By training this inference model to generate reasoning sequences that, when processed through the generative mechanism, reproduce the observed answers, we approximate the underlying computation that the model performs. We simultaneously utilize prior information, construct an EM scoring function, and further guide the model to generate high-scoring inversion CoT.

A distinctive feature of our approach is an explicit alignment objective that connects the inferred reasoning path to the model's own generated CoT. While the explicit CoT may be an imperfect or partially faithful record of the true reasoning steps, it often encodes valuable signals about the intended path. By encouraging similarity between the inferred and explicit reasoning sequences at the level of content or structure, we can promote interpretability and enable direct comparison between what the model claims and what it actually computes. This alignment objective, integrated into the variational learning framework, makes use of the explicit CoT as a soft guiding signal while remaining robust to potential unfaithfulness. Our proposed method, CoT Inversion, recovers the true reasoning dynamics behind language model answers.

Our work makes the following key contributions:

- We develop a probabilistic modeling framework that treats the model's reasoning path as a latent variable linking input and output. Variational inference is used to recover hidden reasoning sequences, guided by an EM scoring function, enabling the separation of authentic reasoning from post-hoc explanations.

- Moreover, we apply the CoT style alignment to our training process to encourage minimal yet sufficient explanations. This leads to more concise and interpretable intermediate representations of reasoning.

- Through comprehensive experiments, our approach consistently improves faithfulness and interpretability in reasoning traces. Results show significant gains on benchmark datasets compared to existing baselines.

## 2 RELATED WORK

**Reasoning Model Faithful** The transparency offered by CoT reasoning in LLMs is appealing for safety and interpretability, allowing potential monitoring of the model's reasoning process. However,

a growing body of work questions the faithfulness of these generated rationales (Baker et al., 2025). Studies indicate that CoT explanations do not always accurately represent the model's internal computations or the factors affecting its final prediction accuracy (Chua & Evans, 2025; Arcuschin et al., 2025; Lanham et al., 2023; Lyu et al., 2023). For instance, models can be biased by input features, such as the order of multiple choice options, yet fail to mention this influence in their CoT, sometimes generating plausible but misleading justifications for incorrect, biased answers. Research evaluating state of the art models found that while CoTs sometimes reveal the use of reasoning hints (Hammoud et al., 2025), the rate is often low, and reinforcement learning shows limited success in improving faithfulness consistently. Furthermore, while monitoring CoT can be effective for detecting misbehavior like score hacking, even allowing weaker models to monitor stronger ones, there's a risk that models under strong optimization pressure might learn to obfuscate their intent within the CoT, limiting the reliability of monitoring (Chen et al., 2025b). This suggests CoT monitoring is a useful but potentially insufficient tool for ensuring model alignment and detecting subtle failures (Shen et al., 2025; Hou et al., 2025).

**Latent Space Reasoning** Traditional LLMs predominantly perform reason within the discrete space of natural language, often using CoT. However, researchers are increasingly exploring the potential of reasoning in continuous latent spaces, arguing that the language space may not be optimal due to verbosity and the difficulty of representing complex planning (Noh et al., 2025; Kong et al., 2025). One approach involves Latent-Thought Language Models (LTMs), which incorporate explicit latent thought vectors that follow a prior distribution and guide token generation via a Transformer decoder (Tang et al., 2025). These models are trained using variational Bayes to infer the posterior distribution of these latent vectors, demonstrating improved sample efficiency and scaling properties compared to standard autoregressive models. Another paradigm (Hao et al., 2024), Coconut (Chain of Continuous Thought), utilizes the LLM's final hidden state as a "continuous thought" representation, feeding it back directly into the model's embedding space without decoding it into a word token. This approach has shown promise in augmenting LLM reasoning capabilities, enabling emergent patterns like breadth-first search by encoding multiple reasoning paths within the continuous state, and outperforming CoT on certain logical tasks requiring backtracking. Furthermore, (Hagendorff & Fabi, 2025) explicitly modeling and inferring latent thoughts is proposed as a way to improve pretraining data efficiency, viewing text as a compressed outcome of a richer thought process (Chen et al., 2025a; Mittal et al., 2024; Geiping et al., 2025).

**Inversion in Language Model** Research on language model inversion has revealed significant privacy and security concerns. (Morris et al., 2023) demonstrated that autoregressive LLMs encode prompt information in their output distributions, making input reconstruction feasible. (Zhang et al., 2024) proposed black-box prompt extraction through output inversion, highlighting the threat even without access to model internals. The study (Geiping et al., 2024) exposed vulnerabilities where attackers manipulate LLMs to disclose confidential prompts, directly showcasing inversion risks. Similarly, (Skapars et al., 2024) analyzed the prospects of exact inversion in the context of slander detection, revealing both practical challenges and the risk of prompt exposure. Finally, research on embedding inversion (Liu et al., 2024) broadened the scope by demonstrating how embedding representations themselves are susceptible to inversion, and proposed mitigation techniques to limit sensitive information leakage. Collectively, these works establish inversion as a serious concern for LLM deployment and motivate ongoing research in mitigation.

## 3 PRELIMINARY

LLMs have demonstrated strong capabilities in complex reasoning, often by generating explicit CoT explanations $c$ alongside their answers $a$ for a given input $x$. However, recent studies suggest that the explicit CoT $c$ produced by the model may not faithfully reflect the *true latent reasoning process* that actually leads to the answer $a$. This raises crucial questions about the alignment and faithfulness between $c$ and the actual model reasoning, which could be implicitly encoded as a latent variable $z$.

**Latent Variable Modeling** We hypothesize that for each input $x$, there exists a latent, possibly unobserved, reasoning process $z$ that mediates between input $x$ and output answer $a$. The CoT $c$ generated by the model is an explicit but potentially imperfect or incomplete explanation. Our modeling objective is to reconstruct the likely latent reasoning chain $z$ that faithfully led to $a$, and to

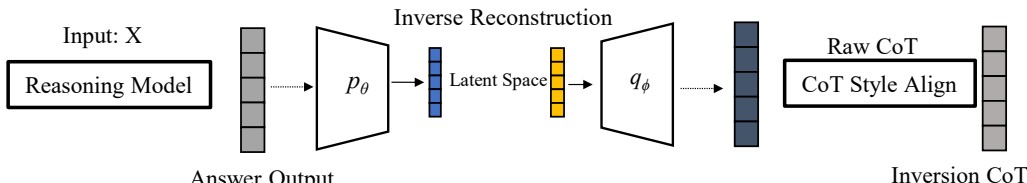

Figure 2: Inverse CoT Framework

contrast it with the explicit CoT $c$ for faithfulness evaluation. By modeling $z$ as a stochastic variable conditioned on both $x$ and $a$, we capture alternative reasoning paths that are consistent with the correct answer. This probabilistic framework allows us to assess how well $c$ approximates the true underlying reasoning, enabling a more nuanced and principled evaluation of explanation quality beyond surface-level similarity.

**Variational Inference (VI)**   In most practical cases, the true posterior $p(z \mid x, a)$ is intractable, especially when $z$ is a complex, structured sequence Blei et al. (2017). We therefore introduce a variational approximation $q_\phi(z \mid x, a)$ parameterized by an inference network. VI provides a tractable surrogate for the log marginal likelihood $\log p(a \mid x)$ via the evidence lower bound (ELBO), encouraging $z$ to be predictive of $a$ while regularizing $q_\phi$ toward a prior $p(z \mid x)$.

**Entropy-Weighted Prior and the CoT Inversion Score**   To ground a discrete search over CoT sequences, we endow the CoT space with a principled prior and an associated score. Let $M$ be a frozen reasoning model used only for *scoring*. While generating the original CoT $c$ with $M$, we record per-token entropies

$$H_i = -\sum_w P_M(w \mid x, c_{<i}) \log P_M(w \mid x, c_{<i}), \qquad \text{cost}(i) = \frac{1}{1 + \alpha H_i}. \tag{1}$$

Using these costs, we define an entropy-weighted edit distance $D_H(\tilde{c}, c)$ and specify a CoT prior

$$\log p(\tilde{c} \mid x) \approx -\lambda D_H(\tilde{c}, c) + C, \tag{2}$$

which favors edits at high-entropy (uncertain) locations and discourages edits at low-entropy (certain) ones. Given this prior and the answer-likelihood under $M$ (computed by forced decoding), the *inversion score* for any candidate $\tilde{c}$ is

$$\text{Score}(\tilde{c}) = \log p_M(a \mid x, \tilde{c}) - \lambda D_H(\tilde{c}, c). \tag{3}$$

This score serves as our EM-style objective in the discrete CoT space.

## 4   METHOD

**Overview**   Figure 2 illustrates the overall architecture of our inverse reconstruction framework, designed to examine the alignment between model-generated answers and interpretable reasoning steps. Specifically, given an initial input $x$, a reasoning model first produces the Answer Output, denoted as $a$. To analyze the underlying reasoning process, we introduce two components: a prior model $p_\theta$ and a reconstruction model $q_\phi$. The prior model $p_\theta$ maps the answer output into a latent space, encoding potential implicit reasoning representations. The reconstruction model $q_\phi$ then decodes these latent representations to recover Inversion CoT. We aim to assess the faithfulness of an explicitly generated CoT $c$ to the final answer $a$ produced by a large language model given an input $x$. Our core idea is to postulate the existence of a *latent* reasoning process $z$ that represents the underlying, potentially unarticulated, steps leading from $x$ to $a$. We then develop a method to infer this latent $z$ and compare it against the observed $c$. A high degree of similarity between the inferred $z$ and the explicit $c$ would suggest faithfulness, while divergence would indicate potential issues like post-hoc rationalization or unstated reasoning shortcuts.

### 4.1 PROBABILISTIC MODELING FRAMEWORK

We formulate the problem within a probabilistic generative framework. We assume that the final answer $a$ is generated based on the input $x$ and the latent reasoning chain $z$. The explicit CoT $c$ is considered an observed variable alongside $x$ and $a$, but it is not assumed to be part of the direct causal path from $x$ to $a$ in our generative model for inferring $z$. The generative process is conceptualized as:

1. An input $x$ potentially gives rise to a latent reasoning path $z$ according to a prior distribution $p(z \mid x)$. This represents the plausible reasoning steps one might expect given only the input.

2. The final answer $a$ is generated conditioned on the latent reasoning $z$ and the input $x$, following a likelihood distribution $p(a \mid z, x)$.

Our primary goal is to compute the posterior $p(z \mid x, a)$, but direct computation is generally intractable.

### 4.2 VARIATIONAL INFERENCE

To overcome the intractability of the posterior, we employ Variational Inference. We introduce a parameterized variational distribution $q_\phi(z \mid x, a)$, implemented by an *inference network* with parameters $\phi$, to approximate the true posterior $p(z \mid x, a)$.

**Theorem 1** (ELBO for CoT Inversion). *The objective of VI is to minimize the Kullback–Leibler (KL) divergence between the approximate posterior $q_\phi(z \mid x, a)$ and the true posterior $p(z \mid x, a)$, which is equivalent to maximizing the ELBO:*

$$\log p(a \mid x) \ \geq \ \mathcal{L}_{ELBO}(\theta, \phi) \ = \ \mathbb{E}_{z \sim q_\phi(z|x,a)}\big[\log p_\theta(a \mid z, x)\big] \ - \ D_{\mathrm{KL}}\big(q_\phi(z \mid x, a) \,\|\, p(z \mid x)\big). \tag{4}$$

*Here, $p_\theta(a \mid z, x)$ is the likelihood function, parameterized by $\theta$ (often a generator/decoder). The first term encourages the inferred $z$ to be predictive of $a$, while the second term regularizes $q_\phi$ toward the prior $p(z \mid x)$.*

For a detailed proof of Theorem 1, please refer to the Appendix.

The reconstruction expectation $\mathbb{E}_{z \sim q_\phi(z|x,a)}[\log p_\theta(a \mid z, x)]$ ensures that the latent $z$ inferred from $(x, a)$ carries the information necessary to yield $a$ from $x$. However, $z$ lives in a continuous space learned indirectly from data, while CoT explanations are discrete sequences with fine-grained, token-level structure. Relying solely on $z$ may conflate many semantically distinct chains that happen to predict $a$. To connect the continuous objective in equation 1 with sequence-level faithfulness, we therefore complement VI with a discrete search over explicit CoTs. This search is guided by an entropy-weighted prior centered at the original CoT $c$ (Eq. equation 2) and a sequence-level score that directly trades off answer likelihood and edit cost (Eq. equation 3). The result is a coupled procedure: VI shapes $z$ for predictive sufficiency, while the discrete search identifies a concrete chain $\tilde{c}$ that is both plausible under the prior and strongly predictive of $a$ under the frozen scorer $M$.

### 4.3 DISCRETE COT INVERSION USING THE ENTROPY-WEIGHTED SCORE

In parallel to latent-space VI, we search in the discrete CoT space for an *Inversion CoT* $\tilde{c}$. The prior over CoTs is given in Eq. equation 2; candidates are ranked by the inversion score in Eq. equation 3, where $p_M(a \mid x, \tilde{c})$ is computed by forced decoding under the frozen scorer $M$. Practically, we implement $q_\phi(\tilde{c})$ as an editor that proposes local edits around $c$ (e.g., with beam search), leading to a coordinate-ascent procedure that alternates discrete updates to $\tilde{c}$ using Eq. equation 3 and continuous updates to $z$ via Eq. equation 4.

To make the connection explicit, note that the CoT-space ELBO for any proposal distribution $q_\phi(\tilde{c})$ satisfies

$$\mathbb{E}_{\tilde{c} \sim q_\phi}\big[\log p_M(a \mid x, \tilde{c})\big] - D_{\mathrm{KL}}\big(q_\phi(\tilde{c}) \,\|\, p(\tilde{c} \mid x)\big) \ = \ \mathbb{E}_{\tilde{c} \sim q_\phi}\big[\mathrm{Score}(\tilde{c})\big] + \mathrm{const}, \tag{5}$$

where the constant depends only on the normalization in Eq. equation 2. Hence maximizing the expected score is equivalent (up to constants) to maximizing a sequence-level ELBO. Moreover, one

---

**Algorithm 1** Inversion CoT

---

**Input:** $x, a, c, M, E_\phi, p_{\text{prior}}(z \mid x), q_\phi(z \mid x, a), p_\theta(a \mid z, x)$

**Output:** $\tilde{c}, \theta, \phi$

 1: Compute $H_i$ on $c$ with $M$; define $\text{cost}(i) = \frac{1}{1+\alpha H_i}$ and $D_H$; fix prior via Eq. equation 2
 2: Initialize $\tilde{c} \leftarrow c$; initialize $z \sim q_\phi(z \mid x, a)$
 3: **for** $t = 1$ **to** $N$ **do**
 4:     Propose candidates $\mathcal{C}$ around $\tilde{c}$ using $E_\phi$ (beam $k$, conditioned on $x, a$)
 5:     $\tilde{c}' \leftarrow \arg\max_{\tilde{c} \in \mathcal{C}} \text{Score}(\tilde{c})$ using Eq. equation 3
 6:     **if** $\text{Score}(\tilde{c}') - \text{Score}(\tilde{c}) < \varepsilon$ **then**
        **break**
 7:     **end if**
 8:     $\tilde{c} \leftarrow \tilde{c}'$; update $(\theta, \phi)$ to increase $\mathcal{L}(\theta, \phi)$ in Eq. equation 4
 9: **end for**
10: Return $\tilde{c}$ and the learned $(\theta, \phi)$

---

can define a joint objective that couples the continuous and discrete parts,

$$\mathcal{J}(\theta, \phi) = \mathcal{L}(\theta, \phi) + \eta \, \mathbb{E}_{\tilde{c} \sim q_\phi}[\text{Score}(\tilde{c})], \qquad \eta > 0, \tag{6}$$

which recovers Eq. equation 4 when $\eta = 0$ and recovers Eq. equation 5 when $\mathcal{L}$ is held fixed. In practice, we alternate (i) proposing and selecting $\tilde{c}$ that improves Eq. equation 3 under the prior in Eq. equation 2, and (ii) updating $(\theta, \phi)$ to improve $\mathcal{L}$, until neither step yields a meaningful improvement. This yields discrete chains that align with token-level uncertainty and continuous representations that remain maximally predictive of the observed answer.

### 4.4 CoT Alignment

This component aligns the inferred latent reasoning $z$ with the explicit CoT chain $c$. We employ a parameterized distribution $p_\psi(z \mid x, c)$ to map input $x$ and the CoT $c$ to the latent space. By minimizing $D_{\text{KL}}\big(q_\phi(z \mid x, a) \,\|\, p_\psi(z \mid x, c)\big)$ (or equivalently maximizing an alignment similarity), we encourage $z$ to capture the structure and semantics present in $c$. During inference without CoTs, only $q_\phi(z \mid x, a)$ (or a variant conditioned on $x$) and $p_\theta(a \mid z, x)$ are used. Given an LLM-generated CoT $c_{LLM}$ and answer $a_{LLM}$ for input $x$, we compute $z_c \sim p_\psi(z \mid x, c_{LLM})$ and $z_a \sim q_\phi(z \mid x, a_{LLM})$ and measure their cosine similarity:

$$\text{Similarity}(z_c, z_a) = \text{Sentence Cosine}(z_c, z_a). \tag{7}$$

A higher value indicates that the explicit chain follows a pathway in latent space consistent with the process optimized for answer generation. We further probe faithfulness by perturbing $z$ and observing $p_\theta(a \mid z_{\text{perturbed}}, x)$; differences between perturbations from $z_c$ versus $z_a$ reveal whether $c_{LLM}$ tracks causal steps that actually drive $a_{LLM}$.

**Theorem 2** (Inversion CoT Answer Fidelity Bound). *Let $z^* = \arg\max_z q(z \mid x, a)$ (i.e., the highest-scoring beam output), and define the true joint posterior $p^*(z \mid x, a) \propto p_\theta(z \mid x) p_\theta(a \mid z, x)$. Let $D = \text{KL}(q(z \mid x, a) \| p^*(z \mid x, a))$. Then,*

$$p_\psi(a \mid z^*) \geq p_\theta(a \mid x) \cdot \frac{\exp(-D)}{p_\theta(z^* \mid x)}. \tag{8}$$

For detailed proof about Theorem 2, please refer to the Appendix.

This probabilistic framework allows us to move beyond surface-level analysis of CoT strings and provides a principled way to quantify faithfulness by grounding it in a latent space optimized for task performance and aligned with available reasoning examples. All Inversion CoT algorithms are as follows in Algorithm 1.

## 5 Experiment

**Setup** Our study rigorously evaluates the faithfulness and interpretability of reasoning language models by examining the causal link between their CoTs and produced answers, asking whether

Table 1: Faith CoT inversion percentage.

| Models | Harmful Benchmark | | Faithful Cases | | | | | |
| | SafeChain | SafeR1 | Num. Labeled Pairs | Switching Arguments | Expert Opinion | Fact Manipulation | Answer Flipping | Other |
|---|---|---|---|---|---|---|---|---|
| Raw CoT | 15.8% | 11.2% | 10.7% | 4.6% | 27.2% | 12.2% | 14.5% | 15.9% |
| Inversion CoT | 2.4% | 1.7% | 3.5 % | 2.7 % | 7.1% | 9.6% | 8.8% | 6.2% |

answers truly follow the reasoning implied by the CoT or diverge from it. For CoT generation, we use QwQ 32B and DeepSeek-R1-Distill-Qwen-32B as open-source reasoning models; these are employed solely to produce CoT traces. To interrogate faithfulness, we invert the reasoning process by reconstructing plausible CoTs directly from answers using Qwen2.5 32B as the inversion model; here, we train only LoRA while keeping the backbone frozen, thereby localizing adaptation to the answer-to-CoT mapping. To preserve stable CoT priors and control for confounding architectural or training effects, a frozen Qwen2.5 3B base is retained as a prior anchor: it remains unchanged and supplies latent constraints that regularize the inversion procedure. Concretely, the inversion head on Qwen2.5 32B is lightweight and task-specific, trained under these priors so that latent knowledge is kept intact and only minimal task adaptation occurs via LoRA. We evaluate the accuracy of the inverted CoTs using GSM8K, MATH500, and AIME24, representing a comprehensive suite of math and symbolic reasoning datasets. For faithfulness and safety evaluation, we employ the SafeChain and Safe R1 datasets.

**Inversion Hint Evaluation**   Table 1 presents a comparative evaluation between Raw CoT and CoT Inversion methods, particularly focusing on the percentage of cases in which Question Hints are used. The table is split into two principal sections: Harmful Benchmark and Faithful Cases. In the Harmful Benchmark section, two tasks, SafeChain Wang et al. (2025b) and SafeR1 Wang et al. (2025a), are analyzed. The percentages indicate how often the models utilized question hints in harmful scenarios, which could potentially lead to unsafe or biased reasoning. For these benchmarks, Raw CoT demonstrates relatively high frequencies of hint usage, with 15.8% for SafeChain and 11.2% for SafeR1. In contrast, after applying CoT Inversion, the rates decrease dramatically to 2.4% and 1.7% respectively. This substantial reduction suggests that CoT Inversion effectively suppresses the model's reliance on question hints in contexts where such behavior may be undesirable or risky.

The Faithful Cases part of the table further divides the evaluation into several more specific reasoning categories. These include the total number of labeled pairs, switching arguments, expert opinion, fact manipulation, answer flipping, and other types of reasoning cases. Across all of these categories, CoT Inversion consistently shows a lower percentage of question hint usage compared to Raw CoT. For example, in the expert opinion category, the usage drops from 27.2% (Raw CoT) to 7.1% (Inversion CoT). Similarly, for fact manipulation, the rates go from 12.2% to 9.6%. Other categories, such as switching arguments (4.6% to 2.7%) and answer flipping (14.5% to 8.8%) reveal analogous improvements. The overall trend indicates that CoT Inversion helps the model become more faithful by reducing its dependence on question hints, resulting in a more transparent and reliable chain of reasoning. In summary, the table illustrates that CoT Inversion consistently enhances both safety and faithfulness across different reasoning tasks by limiting unnecessary or misleading use of question hints compared to the Raw CoT approach.

**Unfaithfulness Percentages**   Table 2 summarizes the experimental results comparing different LLMs setups for the faithfulness of their outputs. The two main variables are the use (raw or inversion) of a debiasing instruction and the model type. Two inversion strategies are tested: zero-shot (ZS) and few-shot (FS), with and without CoT reasoning. The metrics reported are the percentage of unfaithful outputs overall (% Unfaith. Overall) and the percentage of unfaithful explanations specifically caused by bias (% Unfaith. Expl. by Bias). Results are reported in both the No debiasing instruction and the Debiasing instruction conditions. An Unbiased baseline (without model intervention) is also shown.

The results show that the use of debiasing instructions generally reduces the percentage of unfaithful outputs, especially for the Reasoning model. CoT increases unfaithful explanations in some cases, but debiasing mitigates this effect. Inversion CoT often improves faithfulness over zero-shot inversion, particularly for Reasoning models. Overall, model choice, inversion strategy, and explicit debiasing all play important roles in reducing unfaithful and biased model explanations.

Table 2: Unfaithfulness percentages for Reasoning vs No Reasoning models with and without debiasing instructions. ZS: Zero-shot, FS: Few-shot.

| | | % Unfaith. Overall | | % Unfaith. Expl. by Bias | |
|---|---|---|---|---|---|
| | | Raw CoT | Inversion CoT | Raw CoT | Inversion CoT |
| *No debiasing instruction* | | | | | |
| Unbiased | | - | - | 52.3 | 52.4 |
| Reasoning | ZS | 24.3 | 28.6 | 63.1 | 61.5 |
| Reasoning | FS | 19.4 | 25.6 | 62.4 | 58.6 |
| No Reasoning | ZS | 31.8 | 28.2 | 59.4 | 56.7 |
| No Reasoning | FS | 25.3 | 22.9 | 71.0 | 64.6 |
| *Debiasing instruction* | | | | | |
| Reasoning | ZS | 22.3 | 27.4 | 62.0 | 62.4 |
| Reasoning | FS | 17.7 | 24.3 | 63.2 | 54.1 |
| No Reasoning | ZS | 22.4 | 24.6 | 51.1 | 47.9 |
| No Reasoning | FS | 28.2 | 19.6 | 53.9 | 52.8 |

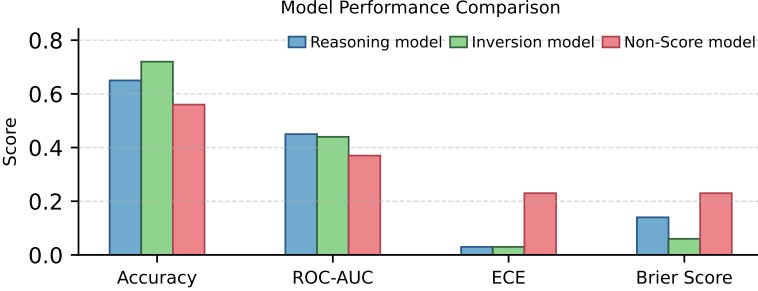

Figure 3: Comparison of Model Performance Across Faithful Evaluation Metrics.

**Inverse CoT Length Effect**  The experimental results in Table 3 clearly demonstrate that Inversion CoT achieves accuracy and reasoning length comparable to the original CoT method with only a slight increase in output tokens. Specifically, on all three datasets GSM8k, MATH500, and AIME24, Inversion CoT produces reasoning outputs that are only marginally longer than those of standard CoT. This indicates that Inversion CoT can replicate native CoT performance with just a minimal increase in reasoning length. Moreover, the variants without style alignment (w/o style align) result in much longer reasoning steps, which illustrates that style alignment is essential. Without it, the model generates unnecessarily verbose and lengthy outputs, reducing its efficiency and coherence. Furthermore, comparisons with simply removing KL loss or using pause tokens further reveal that neither simplification nor minimal interventions produce effective or concise CoT outputs. Therefore, both CoT style alignment and the Inversion CoT framework are necessary: style alignment ensures output efficiency and clarity, while Inversion CoT enables models to match the original CoT performance with only a slight increase in reasoning length, making it a practical and robust approach for complex reasoning tasks.

Table 3: Results on GSM8k, MATH500, and AIME24 datasets. Len. (%) indicates the accuracy based on the proportion of CoT tokens to the total output tokens.

| Method | GSM8k | | MATH500 | | AIME24 | |
|---|---|---|---|---|---|---|
| | Len. (%) | # Tokens | Len. (%) | # Tokens | Len. (%) | # Tokens |
| CoT | 42.9 | 452 | 40.2 | 1530 | 42.9 | 2636 |
| No-CoT | 6.5 | 47 | 6.2 | 55 | 6.5 | 66 |
| Simplify CoT | 30.0 | 178 | 30.1 | 1210 | 30.0 | 1252 |
| Pause token | 1.4 | 13 | 1.2 | 16 | 1.4 | 19 |
| Inversion CoT (Ours) | 49.1 | 517 | 52.1 | 1607 | 51.7 | 2728 |
| - w/o style align | 62.4 | 689 | 60.4 | 1810 | 60.1 | 2972 |
| - w/o KL | 51.6 | 546 | 52.6 | 1642 | 49.6 | 2770 |
| - pause token | 2.1 | 20 | 2.0 | 24 | 1.9 | 29 |

Table 4: Entropy token evaluation.

| Method | GSM8K | | MATH | | AIME24 | |
|---|---|---|---|---|---|---|
| | HET (%) | # Clus. | HET (%) | # Clus. | HET (%) | # Clus. |
| ReAct | 25.2 | 21 | 25.8 | 22 | 26.1 | 22 |
| Self-Consistency (Voting) | 23.5 | 19 | 24.1 | 20 | 24.5 | 21 |
| Tree-of-Thought / Search | 21.8 | 18 | 22.4 | 19 | 22.9 | 20 |
| Skeleton-of-Thought | 21.0 | 15 | 21.7 | 16 | 22.2 | 17 |
| RAP | 20.1 | 12 | 20.9 | 14 | 21.5 | 15 |
| CoT Inversion (Ours) | 14.8 | 5 | 15.1 | 6 | 15.6 | 8 |

**Inverse CoT Score Evaluation**    As Figure 3 shows, the experimental results demonstrate the superior performance of the Inverse Model in comparison to other models, including the Non score model and the Reasoning model. As shown in the metrics of Accuracy, ROC-AUC, and Expected Calibration Error (ECE), the Inverse Model achieves significantly higher scores across all categories. Specifically, it records an Accuracy, ROC-AUC, and ECE that are notably better than the Non-score model's corresponding results. These results indicate that the inverse model exhibits stronger predictive capability and better calibration for estimating probabilities. Moreover, the Brier Score further highlights the effectiveness of the Inverse Model, with a result that stands out compared to the lower-performing models' scores. This suggests that the Inverse Model provides more reliable and accurate probabilistic predictions. The consistent gap in performance across all metrics underscores the advantage of the Inverse Model in capturing complex patterns and improving generalization. Overall, these findings highlight its robustness and potential applicability in real world scenarios where accurate and well-calibrated predictions are critical.

**Low-Entropy Token Evaluation**    Table 4 reports the high-entropy token (HET) statistics across three benchmarks and methods. Baseline approaches (ReAct, Self-Consistency, Tree-of-Thought, Skeleton-of-Thought, and RAP) exhibit HET ratios between 20.1% and 25.2% and generate 12 to 22 distinct HET clusters depending on the dataset and method. By contrast, our CoT Inversion method consistently lowers HET ratios to the 14.8%–15.6% range and collapses cluster counts to 5–8. This reduction in both token-level entropy and cluster diversity indicates a decisive shift toward more constrained and repeatable reasoning vocabularies. Lower HET ratios mean fewer unpredictable tokens appear in CoT outputs, while smaller cluster counts reflect the model converging on a compact set of semantic patterns during reasoning. Together these effects produce deterministic, low-entropy reasoning trajectories that are easier to interpret, debug, and verify. Importantly, this behavior does not merely compress token usage: it preserves the essential structural diversity needed to solve problems across GSM8K, MATH, and AIME24 while removing spurious lexical variation that obscures logical flow. As a consequence, CoT Inversion yields CoT paths that are both semantically coherent and operationally stable, enabling systematic failure analysis and principled refinement of reasoning procedures. Empirically, this leads to improved reproducibility across repeated runs and tighter confidence calibration for downstream selection policies. The compressed reasoning lexicon simplifies human inspection and supports lightweight post-hoc verification tools, reducing the cost and time of manual auditing. These properties make the method particularly attractive for applications that require transparent, auditable, and high-assurance inference.

## 6  CONCLUSION

We introduce CoT Inversion, a method that process reasoning faithfulness in RLLMs by treating true reasoning as a latent variable connecting input to answer and applying variational inference with an EM scoring algorithm. Our method incorporates an explicit alignment objective encouraging similarity between inferred latent CoT and explicitly generated CoT, enabling quantitative measurement of reasoning faithfulness. Experiments demonstrate this framework effectively distinguishes faithful from unfaithful reasoning patterns, providing valuable insights into model behavior, enhancing AI transparency and trustworthiness, and promoting safer deployment of reasoning capable models by identifying when explanations may be post-hoc rationalizations rather than authentic reasoning processes.

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

# A  APPENDIX

## A.1  PROOF OF THEOREM 1

*Proof.* We begin by expanding the log-likelihood of the marginal action distribution:

$$\log p(a|x) = \log \int p(a|z,x)p(z|x)\, dz.$$

This integral is rewritten using importance sampling with the inference model $q_\phi(z|x,a)$:

$$\log p(a|x) = \log \int p(a|z,x)p(z|x)\frac{q_\phi(z|x,a)}{q_\phi(z|x,a)}\, dz = \log \mathbb{E}_{z\sim q_\phi(z|x,a)}\left[\frac{p(a|z,x)p(z|x)}{q_\phi(z|x,a)}\right].$$

Applying Jensen's inequality to move the expectation inside the log yields a lower bound:

$$\log p(a|x) \geq \mathbb{E}_{z\sim q_\phi(z|x,a)}\left[\log \frac{p(a|z,x)p(z|x)}{q_\phi(z|x,a)}\right].$$

Expanding the logarithm inside the expectation gives:

$$\log p(a|x) \geq \mathbb{E}_{z \sim q_\phi(z|x,a)}[\log p(a|z,x)] + \underbrace{\mathbb{E}_{z \sim q_\phi(z|x,a)}[\log p(z|x) - \log q_\phi(z|x,a)]}_{=-\mathrm{KL}(q_\phi(z|x,a)\,||\,p(z|x))}.$$

Putting it all together, we obtain the final variational lower bound:

$$\log p(a|x) \geq \mathbb{E}_{z \sim q_\phi(z|x,a)}[\log p(a|z,x)] - \mathrm{KL}(q_\phi(z|x,a)\,||\,p(z|x)).$$

### A.2 PROOF OF THEOREM 2

*Proof.* By definition of $p^*$, we have

$$p^*(z|x,a) = p_\theta(z|x)p_\theta(a|z).$$

Taking the logarithm gives

$$\log p_\psi(a|z) = \log p^*(z|x,a) + \log p_\theta(a|z) - \log p_\theta(z|x).$$

Since $z^* = \arg\max_z q(z|x,a)$, it follows from Gibbs' variational inequality that

$$\log p^*(z^*|x,a) \geq \mathbb{E}_q[\log p^*(z|x,a)] - D.$$

Combining these results leads to

$$\log p_\psi(a|z^*) \geq \log p_\theta(a|x) - D - \log p_\theta(z^*|x).$$

Exponentiating both sides yields the desired result.

