# OpenReview forum: "Enabling Reasoning Language Models to Reveal Their True Thoughts via CoT Inversion"
_ICLR.cc/2026/Conference — ICLR 2026 Conference Withdrawn Submission_

### Official Review · Reviewer_Pkym · 2025-10-25

**Soundness:** 2
**Presentation:** 1
**Contribution:** 2
**Rating:** 4
**Confidence:** 3

**Summary:**

This paper addresses the critical problem of faithfulness in CoT. The authors argue that the explicit CoT generated by LLM may not reflect the model's actual computational process, but could instead be a post-hoc rationalization. This discrepancy undermines the use of CoT for interpretability and safety monitoring.

To solve this, the authors propose CoT Inversion, a novel probabilistic framework to infer the "true" underlying reasoning. The core idea is to treat the genuine reasoning chain as a latent variable that mediates between an input and the final answer. The method then uses variational inference to "reverse" the generative process and infer this latent chain given the input and the answer.

The authors' goal is to use this framework to create a quantitative metric for CoT faithfulness by comparing the inferred chain with the explicit one. Experiments on safety and reasoning benchmarks suggest the method can identify unfaithful reasoning patterns and produces low-entropy reasoning paths.

**Strengths:**

Novel Problem Formulation: The paper tackles one of the most significant and popular open questions in LLM safety and interpretability: is the model's CoT explanation faithful to its internal reasoning process? The approach of inverting the process to infer the reasoning from the answer $p(z|x, a)$ is a original and powerful framing of the problem, moving beyond simply monitoring the provided CoT.

Methodological Contribution: The "entropy-weighted edit distance" is a novel algorithm contribution. It provides a principled, probabilistic prior for the discrete search space, formalizing the intuition that a faithful reasoning chain should only deviate from an unfaithful one at points of high uncertainty.

**Weaknesses:**

Clarity Issue in Methodology:
1. Unclear definition of parameters: In section 4, Overview, the author defines $p_{\theta}$ as answer $\rightarrow$ z and $𝑞_{\phi}$ as z $\rightarrow$ CoT. However, in section 4.2, the author defines the likelihood as $p_{theta} (a | z,x)$, which means the $\theta$ maps latent z and input x $\rightarrow$ z.
2. Theorem 2 Bound: It can exceed 1 when $p_{\theta}(z^*|x)$is small. Looking at the theorem 2 proof, I'm a bit confused how the step 2 transformed from step 1. If we substitute the definition from step 1 into step 2, we get: $log p_{\psi}(a|z) = [log p_{\theta}(z|x) + log p_{\theta}(a|z)] + log p_{\theta}(a|z) - log p_{\theta}(z|x)$, so we end up: $log p_{\psi}(a|z) = 2 \cdot log p_{\theta}(a|z)$.

Specifying the discrete editor:
Algorithm 1 relies on an editor 𝐸 "proposing candidates (beam k) around c" and a coordinate-ascent loop, but details are missing.

Interpretation of Experiment Results:
1. The authors claim that style alignment is "essential" and "ensures output efficiency and clarity". However, the data in Table 3 shows that the "w/o style align" ablation achieves dramatically higher accuracy than the final "Inversion CoT (Ours)" model (e.g., 62.4% vs. 49.1% on GSM8k). This indicates that the style alignment, while making the output more concise, acts as a regularizer that significantly harms task performance. This is a critical trade-off, not a simple "essential" component for success.
2. Ablations can be enhanced: there’s "w/o style align", "w/o KL", "pause token", but we need: (i) no discrete inversion, (ii) no VI (score search only), (iii) swap out the entropy weighting for a plain Levenshtein prior.
3. Baselines for "faithfulness" should include self-consistency, self-verification, process supervision/critic models, and recent latent-reasoning or faithfulness probes, evaluated under the same measurement. Current comparisons (e.g., ReAct, ToT) in Table 4 do not clearly measure faithfulness versus diversity/entropy.

**Questions:**

1. Can the authors please clarify the fundamental contradiction in the notation?
2. Why does the "Inversion CoT" method result in a higher "% Unfaith. Overall" in Table 2  compared to the "Raw CoT" baseline in most settings? This result seems to invalidate the central claim of the paper, yet it is not discussed.
3. What, precisely, is the "Inversion CoT" that is evaluated in Tables 1, 2, and 3? Is it the discrete chain found via the EM-like search (Sec 4.3) , or is it a chain generated by decoding the continuous latent variable z (Sec 4.2)?
4. In Table 3, the "w/o style align" model is 13.3% more accurate on GSM8k than the final proposed model. Does this not suggest that aligning the inferred reasoning z with the explicit CoT c actually harms the model's ability to find the correct answer?

---

### Official Review · Reviewer_vsTH · 2025-10-30

**Soundness:** 2
**Presentation:** 2
**Contribution:** 2
**Rating:** 2
**Confidence:** 3

**Summary:**

This paper introduces a novel approach called CoT Inversion, designed to evaluate the faithfulness of generated Chain-of-Thought (CoT) reasoning processes. The method falls under the framework of the variational information bottleneck (VIB), in which separate encoder and decoder networks are trained. Beyond its interpretability perspective, CoT Inversion can also enhance CoT reasoning performance by aligning the inferred latent CoT representations with the generated CoT through a similarity-based optimization objective. While the underlying idea is intuitive, the practical implementation pipeline is relatively complex.

**Strengths:**

The core idea behind the proposed method is intuitive: the entire reasoning process should be abstracted and represented within a latent space, and this representation should be learned through the variational information bottleneck framework.

**Weaknesses:**

- The paper is difficult to follow. For example:

  - Methodology: The entropy-weighted edit distance is not clearly formulated, and several notations are undefined (e.g., ( w ) in Equation 1). It is also unclear how ( p(\tilde{c} \mid x) ) is computed.
  - Experiments: The implementation details of the CoT Inversion method used for comparison against the Raw CoT method are not well described.
  - Overall pipeline: The general workflow of the proposed approach is hard to grasp, and Figure 2 is too abstract to effectively aid understanding.
- The paper makes an intuitive but strong assumption—that the causal reasoning process in discrete space can be represented by a single latent vector. Such a claim requires comprehensive theoretical justification and empirical validation, which are lacking in the current work.
- The reported performance improvements are marginal and inconsistent, as shown in Table 2 and Figure 3.
- The evaluation metric is problematic. In Table 3, the authors report “accuracy based on the proportion of CoT tokens to total output tokens,” but do not include the standard accuracy metric for comparison. Furthermore, this custom metric should be formally defined.

**Questions:**

Please see questions in the weaknesses section.

---

### Official Review · Reviewer_UFn1 · 2025-10-30

**Soundness:** 2
**Presentation:** 2
**Contribution:** 3
**Rating:** 4
**Confidence:** 3

**Summary:**

The paper treats the model’s “true” reasoning process as a latent variable and infer it via variational inference. The inferred latent reasoning is then compared with the model’s explicit CoT rationales.

**Strengths:**

1. The paper addresses a genuinely important and timely concern: that CoT explanations may not faithfully reflect the reasoning used to produce final answers and thus it is dangerous to use them for interpretation or safety monitoring.

2. The proposed probabilistic latent-variable formulation and use of variational inference (with an EM-style objective and entropy-weighted prior) are technically interesting and plausible.

**Weaknesses:**

The main weakness lies in the experimental setup and clarity of evaluation. The descriptions are often too brief or incomplete to fully understand what is being measured.
1. The SafeChain and SafeR1 datasets should be described in greater detail. It is unclear what types of prompts or reasoning tasks they contain, or how they are used to assess CoT faithfulness. Providing a few representative examples would make the evaluation setup much clearer.
2. How is faithfulness judged? Does it come from human annotation, automatic overlap detection, or pre-existing SafeChain metadata?
3. What datasets are Table 2 evaluated on? Are they the same as in Table 1?
4. In Figure 3, the evaluation metrics are not defined or clearly connected to faithfulness. Also, what is the non-score model?

**Questions:**

1. How is the prior $p(z|x)$ chosen?
2. Can you elaborate on why a discrete search over CoT sequences is required, and why it is centered around the original CoT, even though the paper assumes that the original CoT may be unfaithful?

**Details Of Ethics Concerns:**

The paper acknowledges potential privacy risks of model inversion in related work but does not address whether CoT Inversion introduces similar vulnerabilities or how these might be mitigated.

---

### Official Review · Reviewer_ay8b · 2025-11-02

**Soundness:** 3
**Presentation:** 2
**Contribution:** 3
**Rating:** 6
**Confidence:** 3

**Summary:**

This paper proposed a method for evaluating CoT faithfulness via CoT inversion. Specifically, the authors regarded the true reasoning chain as a latent variable that intervened between the input and the answer. They used variational inference to approximate this latent variable by reversing the model’s answer generation process. The faithfulness of the model-generated CoT was then assessed by measuring its similarity to the inferred CoT.

**Strengths:**

$\bullet$  The authors used variational inference to infer the posterior distribution over reasoning chains, effectively inverting the model’s generation process given the input and the answer.

$\bullet$ The method included token-level uncertainty to guide the inversion process.

**Weaknesses:**

$\bullet$  The paper assumes that the recovered reasoning chain should satisfy two criteria:

(1) it should reconstruct the original answer as faithfully as possible;

(2) it should remain close to the original CoT at low-uncertainty tokens (as stated in Eq. (3)).

However, it is unclear why an ideal reasoning chain must satisfy criteria (2). Do the authors implicitly assume that the true CoT is the one with minimal token-level entropy among all CoTs that recover the answer? The paper should first clarify what properties an ideal CoT is expected to possess, and then justify the rationale and limitations of the proposed CoT inversion score. For example, if the true posterior lies outside the editable neighborhood, the inverted chain may not reach the ground truth. Furthermore, allowing large edits on uncertain regions might inadvertently rationalize incorrect generations.
In addition, the paper does not define the entropy-weighted edit distance$D_H(\tilde{c}, c)$ in Eq.(2).

$\bullet$ The authors should explicitly clarify the notations and physical meaning in Theorem 2, which measures the fidelity of the inverted CoT.

$\bullet$  The paper should provide visualizations comparing the recovered CoT and the original CoT, either in the main text or in the appendix, to help readers assess the reconstruction quality.

$\bullet$ The authors should discuss the limitations of using this method to evaluate CoT faithfulness. For example, how does the method depend on the quality of the original model, the score model, and the variational model? Could this lead to self-justifying cycles? If the true posterior is outside the edit neighborhood, can the search process still locate it?

Minor Comments:

$\bullet$ The derivation of Eq. (5) should be included in the appendix.

**Questions:**

In Eq. (6), the purpose of each loss term should be made clearer, i.e., why are both continuous and discrete losses optimized, and what is the corresponding optimization strategy?

---

### Note · Authors · 2025-12-31

I have read and agree with the venue's withdrawal policy on behalf of myself and my co-authors.